# GABA_A_ and GABA_B_ Receptors Mediate GABA-Induced Intracellular Ca^2+^ Signals in Human Brain Microvascular Endothelial Cells

**DOI:** 10.3390/cells11233860

**Published:** 2022-11-30

**Authors:** Sharon Negri, Francesca Scolari, Mauro Vismara, Valentina Brunetti, Pawan Faris, Giulia Terribile, Giulio Sancini, Roberto Berra-Romani, Francesco Moccia

**Affiliations:** 1Laboratory of General Physiology, Department of Biology and Biotechnology “L. Spallanzani”, University of Pavia, 27100 Pavia, Italy; 2Institute of Molecular Genetics (IGM)-CNR “Luigi Luca Cavalli-Sforza”, 27100 Pavia, Italy; 3Laboratory of Biochemistry, Department of Biology and Biotechnology “L. Spallanzani”, University of Pavia, 27100 Pavia, Italy; 4School of Medicine and Surgery, University of Milano-Bicocca, 20900 Monza, Italy; 5Nanomedicine Center, Neuroscience Center, School of Medicine and Surgery, University of Milano-Bicocca, 20900 Monza, Italy; 6Department of Biomedicine, School of Medicine, Benemérita Universidad Autónoma de Puebla, Puebla 74325, Mexico

**Keywords:** hCMEC/D3 cells, GABA, GABA_A_ receptors, GABA_B_ receptors, Ca^2+^ signaling, InsP_3_ receptors, two-pore channels, store-operated Ca^2+^ entry

## Abstract

Numerous studies recently showed that the inhibitory neurotransmitter, γ-aminobutyric acid (GABA), can stimulate cerebral angiogenesis and promote neurovascular coupling by activating the ionotropic GABA_A_ receptors on cerebrovascular endothelial cells, whereas the endothelial role of the metabotropic GABA_B_ receptors is still unknown. Preliminary evidence showed that GABA_A_ receptor stimulation can induce an increase in endothelial Ca^2+^ levels, but the underlying signaling pathway remains to be fully unraveled. In the present investigation, we found that GABA evoked a biphasic elevation in [Ca^2+^]_i_ that was initiated by inositol-1,4,5-trisphosphate- and nicotinic acid adenine dinucleotide phosphate-dependent Ca^2+^ release from neutral and acidic Ca^2+^ stores, respectively, and sustained by store-operated Ca^2+^ entry. GABA_A_ and GABA_B_ receptors were both required to trigger the endothelial Ca^2+^ response. Unexpectedly, we found that the GABA_A_ receptors signal in a flux-independent manner via the metabotropic GABA_B_ receptors. Likewise, the full Ca^2+^ response to GABA_B_ receptors requires functional GABA_A_ receptors. This study, therefore, sheds novel light on the molecular mechanisms by which GABA controls endothelial signaling at the neurovascular unit.

## 1. Introduction

The neurotransmitter γ-aminobutyric acid (GABA) is not only crucial to maintain a proper excitatory:inhibitory balance in neuronal networks by inhibiting glutamatergic pyramidal neurons [1,2,3], but is also instrumental to generate synchronized network oscillations that underpin optimal cognitive functions [4,5]. GABA dampens neuronal excitability by gating Cl^−^-permeable GABA_A_ receptors, thereby enabling Cl^−^ influx down the electrochemical gradient and hyperpolarizing the membrane potential [4]. GABA_A_ receptors typically consist of pentameric channels formed by the combination of three distinct subunits, according to the following stoichiometry: 2α:2β:1γ [3,6]. GABA-dependent inhibition of neurotransmitter release and neuronal activity may also be mediated by metabotropic G_i/o_ protein–coupled GABA_B_ receptors, which are widely distributed in the central nervous system [7,8]. GABA_B_ receptors are composed of an obligatory heterodimer of the GABA_B1_ and GABA_B2_ subunits and induce slow postsynaptic inhibitory potentials by stimulating G_i/o_ protein-coupled inward-rectifying K^+^ channels (GIRK) [7,8]. In addition, it has been documented that GABA_B_ receptors can induce an increase in intracellular Ca^2+^ concentrations ([Ca^2+^]_i_) in rat cortical neurons [9] and mouse cerebellar granule neurons [10]. GABA_B_ receptors trigger intracellular Ca^2+^ signals via a signaling pathway that utilizes heterotrimeric G_i/o_ proteins to stimulate phospholipase Cβ (PLCβ) and cleave phosphatidylinositol 4,5-bisphosphate (PIP_2_) into the two intracellular second messengers inositol-1,4,5-trisphosphate (InsP_3_) and diacylglycerol (DAG). InsP_3_, in turn, binds to InsP_3_ receptors (InsP_3_Rs), which are located on the endoplasmic reticulum (ER), and promotes Ca^2+^ release into the cytoplasm. The following reduction in ER Ca^2+^ levels activates a Ca^2+^ entry pathway that resides on the plasma membrane, known as store-operated Ca^2+^ entry (SOCE), which prolongs over time the Ca^2+^ response to GABA_B_ receptor stimulation [9,10,11].

GABAergic signaling within the neurovascular unit (NVU) has also been described in non-neuronal cells, including astrocytes [12,13] and cerebrovascular endothelial cells [14]. Cortical microvessels, as well as perivascular astrocytes, receive an extensive GABAergic input from local GABA interneurons [15]. Moreover, it has been clearly demonstrated that, during brain development, the pre-formed vascular network guides the tangential journey of GABAergic neurons from the dorsal to the basal telencephalon [16]. Thus, GABAergic neurons can establish a bidirectional communication with cerebrovascular endothelial cells to effectively coordinate these neurovascular interactions [16,17]. Mouse brain microvascular endothelial cells are endowed with the α1, α2, α6, β1, β2, β3, γ1, γ2 and γ3 subunits of GABA_A_ receptors [18], while only functional evidence has been documented in favor of GABA_B_ receptor expression [19]. The role played by endothelial GABA_A_ receptors at the NVU has gathered growing interest upon the discovery that they control cerebral angiogenesis [17] and guide the radial migration of GABA interneurons [16] during embryonic development in mice, and regulate cerebral blood flow (CBF) and prevent neurological deficits in the adult [20]. The signaling pathways that are activated downstream of GABA_A_ receptors in cerebrovascular endothelial cells are yet to be fully unraveled [21]. Preliminary evidence showed that muscimol, a selective GABA_A_ receptor agonist, induced an inward Cl^−^ current and a transient increase in [Ca^2+^]_i_ in mice cerebrovascular endothelial cells [17]. Nevertheless, it is unclear how the inward (i.e., hyperpolarizing) current carried by the ionotropic GABA_A_ receptors could elevate the [Ca^2+^]_i_ in GABA-stimulated cells. Of note, recent studies have revealed that GABA_A_ receptors may trigger a metabotropic (i.e., flux-independent) signaling pathway that leads to InsP_3_-induced Ca^2+^ release from the ER [22,23,24]. Since an increase in endothelial [Ca^2+^]_i_ regulates both angiogenesis [25,26] and CBF [27,28], deciphering the mechanisms whereby GABA induces intracellular Ca^2+^ signals is mandatory to understand the molecular interactions between inhibitory interneurons and adjacent microvessels.

The hCMEC/D3 cell line represents the most suitable in vitro model of human cerebral microvascular endothelial cells [29,30] and has been largely exploited to unveil how cerebrovascular endothelium perceives and transduces neural activity with an increase in [Ca^2+^]_i_ [27,31]. For instance, acetylcholine [32], adenosine trisphosphate (ATP) [33,34], glutamate [35], and histamine [36] bind to their specific G_q_-protein coupled receptors (G_q_PCRs) and stimulate PLCβ to trigger InsP_3_-dependent Ca^2+^ release from the ER. This initial Ca^2+^ peak can be supported by lysosomal Ca^2+^ mobilization through two-pore channels 1 and 2 (respectively, TPC1 and TPC2) and is maintained over time by SOCE activation [27,31]. Intriguingly, the ionotropic N-methyl-D-aspartate (NMDA) receptors (NMDARs) were recently shown to signal an increase in [Ca^2+^]_i_ in hCMEC/D3 cells in a flux-independent mode by interacting with metabotropic glutamate receptors [37]. Herein, we adopted an array of approaches, ranging from real-time quantitative reverse transcription PCR (qRT-PCR) to single-cell Fura-2 imaging to investigate the mechanisms whereby GABA elicits intracellular Ca^2+^ signals in hCMEC/D3 cells. We found that both GABA_A_ and GABA_B_ receptors are expressed and that GABA triggers a biphasic increase in [Ca^2+^]_i_. Pharmacological manipulation revealed that GABA-induced intracellular Ca^2+^ release was mediated by ER Ca^2+^ mobilization through InsP_3_Rs and lysosomal Ca^2+^ discharge via TPC1-2, whereas Ca^2+^ entry was mediated by SOCE. However, GABA_A_ receptors did not mediate any detectable inward current, but they rather signaled in a flux-independent manner to cause intracellular Ca^2+^ release. Conversely, the selective stimulation of GABA_B_ receptors induced the expected metabotropic Ca^2+^ signal that has been described in other cell types. Intriguingly, we provide evidence that GABA_A_ and GABA_B_ receptors interact to elicit a full Ca^2+^ response to GABA in hCMEC/D3 cells. These data provide the first elucidation of the signaling pathways whereby GABA can increase the [Ca^2+^]_i_ in cerebrovascular endothelial cells.

## 2. Materials and Methods

### 2.1. Cell Culture

Human cerebral microvascular endothelial cells (hCMEC/D3) were obtained from the Institut National de la Santé et de la Recherche Médicale (INSERM, Paris, France). hCMEC/D3 cells cultured between passage 25 and 35 were used. As described in [38], the cells were seeded at a concentration of 27.000 cells/cm^2^ and grown in tissue culture flasks coated with 0.1 mg/mL rat tail collagen type 1, in the following medium: EBM-2 medium (Lonza, Basel, Switzerland) supplemented with 5% fetal bovine serum, 1% penicillin–streptomycin, 1.4 μM hydrocortisone, 5 μg/mL ascorbic acid, 1/100 chemically defined lipid concentrate (Life Technologies, Milan, Italy), 10 mM HEPES and 1 ng/mL basic fibroblast growth factor. The cells were cultured at 37 °C, 5% CO_2_ saturated humidity.

### 2.2. Solutions

Physiological salt solution (PSS) had the following composition (in mM): 150 NaCl, 6 KCl, 1.5 CaCl_2_, 1 MgCl_2_, 10 glucose, 10 HEPES. In Ca^2+^-free solution (0Ca^2+^), Ca^2+^ was substituted with 2 mM NaCl, and 0.5 mM EGTA was added. Solutions were titrated to pH 7.4 with NaOH. The osmolality of PSS as measured with an osmometer (Wescor 5500, Logan, UT, USA) was 300–310 mOsm/L.

### 2.3. [Ca^2+^]_i_ Imaging

We utilized the Ca^2+^ imaging set-up that we have described elsewhere [32]. hCMEC/D3 cells were loaded with 4 µM Fura-2 acetoxymethyl ester (Fura-2/AM; 1 mM stock in dimethyl sulfoxide) in PSS for 30 min at 37 °C and 5% CO_2_ saturated humidity. After washing in PSS, the coverslip was fixed to the bottom of a Petri dish and the cells were observed by an upright epifluorescence Axiolab microscope (Carl Zeiss, Oberkochen, Germany), usually equipped with a Zeiss ×40 Achroplan objective (water-immersion, 2.0 mm working distance, 0.9 numerical aperture). The cells were excited alternately at 340 and 380 nm, and the emitted light was detected at 510 nm. A neutral density filter (1 or 0.3 optical density) reduced the overall intensity of the excitation light, and a second neutral density filter (optical density = 0.3) was coupled to the 380 nm filter to approach the intensity of the 340 nm light. A round diaphragm was used to increase the contrast. The excitation filters were mounted on a filter wheel (Lambda 10, Sutter Instrument, Novato, CA, USA). Custom software, working in the LINUX environment, was used to drive the camera (Extended-ISIS Camera, Photonic Science, Millham, UK) and the filter wheel, and to measure and plot on-line the fluorescence from 15–25 rectangular “regions of interest” (ROI) enclosing a corresponding number of single cells. Each ROI was identified by a number. Adjacent ROIs never superimposed. [Ca^2+^]_i_ was monitored by measuring, for each ROI, the ratio of the mean fluorescence emitted at 510 nm when exciting alternatively at 340 and 380 nm [Ratio (F_340_/F_380_)]. An increase in [Ca^2+^]_i_ causes an increase in the ratio [39,40]. Ratio measurements were performed and plotted on-line every 3 s. The experiments were performed at room temperature (22 °C).

### 2.4. Real-Time Reverse Transcription Quantitative PCR (qRT-PCR)

Total RNA was isolated from hCMEC/D3 cells (passage 33) using Trizol reagent (Thermo Fisher Scientific, Milan, Italy) according to the manufacturer’s instructions. After DNAse treatment (Turbo DNA-free™ kit, Thermo Fisher Scientific, Milan, Italy), RNA was quantified using a BioPhotometer D30 (Eppendorf, Hamburg, Germany). For cDNA synthesis, 100 ng RNA was reverse transcribed into 20 µL total volume using the iScriptTM cDNA Synthesis Kit (Bio-Rad, Hercules, CA, USA). Real-time reverse transcription quantitative PCRs with specific primers, designed on an exon–intron junction using the NCBI Primer tool (https://www.ncbi.nlm.nih.gov/tools/primer-blast/ (accessed on 3 March 2022)) [41] (Appendix A) were performed using SsoFast™ EvaGreen^®^ Supermix (Bio-Rad) on a CFX Connect Real-Time System (Bio-Rad) programmed as following: an initial step of 95 °C for 30 s, 40 cycles of 5 s at 95 °C, 5 s at 58 °C. A fluorescence reading was made at the end of each extension step. The PCR mixture consisted of 10 µL SsoFast™ EvaGreen^®^ Supermix (Bio-Rad), 7 µL nuclease-free water, 1 µL cDNA and 1 µL of each forward and reverse primer. All primers were validated by melting curve analyses after each qRT-PCR run and determination of their efficiencies with at least four different cDNA concentrations. Gene expression was evaluated using the ΔΔCt method [42,43]. The genes actin beta (NM_001101) and glyceraldehyde-3-phosphate dehydrogenase (GAPDH) (M17851) were used as endogenous reference for normalizing target mRNA [44]. For each sample, the 2^−ΔΔCt^ value was calculated and represents the gene expression fold-change normalized to the reference gene and relative to the internal calibrator. Data are represented as mean ± SEM of fold-change values. Statistical analysis was performed using log-transformed values of the raw 2^−ΔΔCt^ data. The statistical comparison of fold-changes in gene expression was analyzed with a one-way analysis of variance (ANOVA) followed by Bonferroni’s post-hoc test.

### 2.5. SDS-PAGE and Immunoblotting

hCMEC/D3 cells were lysed with Lysis Buffer (50 mM Tris–HCl, 150 mM NaCl, 1% Nonidet P-40, 1 mM EDTA, 0.25% deoxycholic acid, 0.1% SDS, pH 7.4, 1 mM PMSF, 5 μg/mL leupeptin, and 5 μg/mL aprotinin). Then, 10 μg of total cell proteins was separated by SDS-PAGE and transferred to a PVDF membrane. Membrane probing was performed using the different antibodies diluted 1:1000 in TBS (20 mM Tris, 500 mM NaCl, pH 7.5) containing 5% BSA and 0.1% Tween-20 in combination with the appropriate HRP-conjugated secondary antibodies (1:2000 in PBS plus 0.1% Tween-20). The following antibodies were used: anti-GABA A Receptor α1, clone 3H10 (ZRB1626) from Sigma-Aldrich (Saint Louis, MO, USA), anti-GABA B Receptor 1, clone 2D7 (ab55051) from Abcam (Cambridge, UK). The chemiluminescence reaction was performed using Immobilon Western (Millipore) and images were acquired by the Chemidoc XRS (Bio-Rad, Segrate, MI, Italy).

### 2.6. Electrophysiological Recordings

The presence of GABA_A_-receptor-mediated Cl^−^ currents was assessed by using a port-a-patch planar patch-clamp system (Nanion Technologies, Munich, Germany) in the whole-cell, voltage-clamp configuration, at room temperature (22 °C), as described in [45]. Cultured cells (2–3 days after plating) were detached with Detachin and suspended at a cell density of 1–5 × 10^6^ cells/mL in an external recording solution containing (in mM): 145 NaCl, 2.8 KCl, 2 MgCl_2_, 10 CaCl_2_, 10 HEPES, 10 D-glucose (pH = 7.4). Suspended cells were placed on the NPC© chip surface, and the whole cell configuration was achieved. Internal recording solution, containing (in mM) 10 CsCl, 110 CsF, 10 NaCl, 10 HEPES, 10 EGTA (pH = 7.2, adjusted with CsOH), was deposited in recording chips, having resistances of 3–5 MΩ. To test the ability of GABA_A_ receptors to conduct inward Cl^−^ currents, 100 µM GABA or 30 µM muscimol was added to the external solution. The bioelectrical response to agonist stimulation was recorded in the voltage-clamp mode at a holding potential of −70 mV, as described in [17], by using an EPC-10 patch-clamp amplifier (HEKA, Munich, Germany). Immediately after the whole-cell configuration was established, the cell capacitance and the series resistances (<10 MΩ) were measured. During the recordings, these two parameters were measured, and if exceeding ≥10% with respect to the initial value, the experiment was discontinued [45]. Liquid junction potential and capacitive currents were cancelled using the automatic compensation of the EPC-10 [46]. Data were filtered at 10 kHz and sampled at 5 kHz.

### 2.7. Statistical Analysis of Ca^2+^ Signals

All the data have been obtained from hCMEC/D3 cells from at least three independent experiments. The amplitude of agonist-evoked Ca^2+^ signals was measured as the difference between the ratio at the Ca^2+^ peak and the mean ratio of 30 s baseline before the peak. Pooled data are given as mean ± SEM, while the number of cells analyzed is indicated above the corresponding histogram bars (number of responding cells/total number of analyzed cells). Comparisons between the two groups were done using the Student’s *t*-test, whereas multiple comparisons were performed using ANOVA with the Bonferroni and Dunnett’s post-hoc test, as appropriate. The Bonferroni post-hoc test was used to evaluate multiple comparisons between different means, while Dunnett’s post-hoc test was used to compare each mean to a control mean. *p*-values less than 0.05 were considered statistically significant.

### 2.8. Chemicals

Fura-2/AM was purchased from Molecular Probes (Molecular Probes Europe BV, Leiden, The Netherlands). Nigericin, NED-19, NED-K, and GABA were obtained from Tocris (Bristol, UK). BTP-2 was purchased from Merck Millipore (Darmstadt, Germany). All the other chemicals were of analytical grade and obtained from Sigma Chemical Co. (St. Louis, MO, USA).

## 3. Results

### 3.1. GABA_A_ and GABA_B_ Receptors Are Expressed in hCMEC/D3 Cells

A thorough qRT-PCR analysis was carried out to assess whether and which GABA receptor subunits are expressed in hCMEC/D3 cells, as previously shown in mouse brain cerebrovascular endothelial cells [17,18], by using the specific primers reported in Appendix A. The transcripts encoding for the following GABA_A_ receptor subunits were expressed: α1, α5, β1, β2, and γ1 (Figure 1A). Comparison of the mean fold-change values revealed the following mRNA expression profile: α1 > β1 > α5 = β2 = γ1 (Figure 1A). GABA_B1_ and GABA_B2_ subunit mRNAs were also expressed (Figure 1B), although the transcripts encoding for the GABA_B1_ isoform were significantly up-regulated as compared to GABA_B2_ (Figure 1B). Immunoblotting confirmed that both GABA_A_ α1 and GABA_B1_ subunits were expressed at the protein level (Figure 1C). These data, therefore, demonstrate that GABA receptors are expressed also in the human cerebrovascular endothelial cell line, hCMEC/D3.

### 3.2. GABA Induces a Dose-Dependent Increase in [Ca^2+^]_i_ in hCMEC/D3 Cells

In order to assess whether GABA was able to increase the [Ca^2+^]_i_, hCMEC/D3 cells were loaded with the Ca^2+^-sensitive fluorophore, Fura-2/AM, as described in [35,37]. GABA was found to induce an elevation in [Ca^2+^]_i_ already at a dose as low as 1 pM (Figure 2A). The Ca^2+^ response was consistently observed by challenging hCMEC/D3 cells with increasing concentrations of GABA, ranging from 1 pM to 100 µM (Figure 2A,B). At each dose tested, the Ca^2+^ signal comprised an initial Ca^2+^ peak which decayed to a plateau level that was maintained as long as the agonist was presented to the cells, as shown for the Ca^2+^ response to 100 µM GABA (Figure 2C). Figure 2C shows that, after a short washout, hCMEC/D3 cells were able to promptly respond to a second application of 100 µM GABA. The dose–response relationship did not show the typical S-shaped curve that is mediated by membrane receptors coupled to PLC; accordingly, the amplitude of the initial Ca^2+^ transient was relatively constant between 1 pM and 100 nM (Figure 2B). The highest Ca^2+^ response was elicited by 1 µM GABA, although a robust increase in [Ca^2+^]_i_ could also be observed at higher agonist concentrations, i.e., 10 and 100 µM (Figure 2A,B). Nevertheless, 100 µM is the concentration range that has been exploited by most of the studies investigating how GABA induced intracellular Ca^2+^ signals in neurons and astrocytes [9,10,11,12,13]. Furthermore, GABA concentrations at the synaptic cleft can increase up to 80 µM during neuronal activity [47], although they can reach over-saturating levels (≈3 mM) in response to tetanic stimulation [48]. Therefore, 100 µM GABA, which reliably induces robust elevations in [Ca^2+^]_i_ in hCMEC/D3 cells, was chosen to characterize the underlying signaling pathways.

### 3.3. GABA-Induced Intracellular Ca^2+^ Signals Are Sustained by Extracellular Ca^2+^ Entry through the SOCE Pathway in hCMEC/D3 Cells

The endothelial Ca^2+^ response to extracellular autacoids can be shaped by extracellular Ca^2+^ entry through the plasma membrane and intracellular Ca^2+^ mobilization from endogenous organelles [26,49,50], as also demonstrated in hCMEC/D3 cells [32,33,34,35,36,51]. Therefore, in order to disentangle the contribution of intra- vs. extracellular Ca^2+^ sources to GABA-induced intracellular Ca^2+^ signals, we stimulated the cells in the absence of extracellular Ca^2+^ (0Ca^2+^). Figure 3A–C show that the removal of external Ca^2+^ significantly (*p* < 0.0001) reduced the amplitude of the initial Ca^2+^ peak and abolished the plateau phase, thereby turning the biphasic elevation in [Ca^2+^]_i_ into a transient Ca^2+^ signal. The subsequent restitution of Ca^2+^ to the perfusate induced a second elevation in [Ca^2+^]_i_, which was due to extracellular Ca^2+^ entry (Figure 3B). GABA was removed from the bath 100 s before reintroducing Ca^2+^ into the external saline to prevent the activation of second messenger-operated channels (SMOCs). As discussed elsewhere [32,35,38], the main physiological stimulus coupling GABA receptor activation to extracellular Ca^2+^ influx is, therefore, represented by the initial depletion of the endogenous Ca^2+^ reservoir. In order to confirm that SOCE sustains GABA-induced Ca^2+^ entry, hCMEC/D3 cells were pre-treated with either Pyr6 (10 µM), BTP-2 (20 µM) or Gd^3+^ (10 µM), three distinct inhibitors of Orai1 [35,52,53,54], i.e., the Ca^2+^-selective channel that is activated upon ER Ca^2+^ depletion in these cells [32]. The pharmacological blockade of SOCE with Pyr6 or BTP-2 suppressed GABA-evoked extracellular Ca^2+^ influx in most cells and significantly reduced the amplitude of the residual Ca^2+^ entry in the remaining ones (Figure 3D,E). In accord, 10 µM Gd^3+^, which plugs the pore of Orai1 channels [55,56], fully abolished GABA-evoked extracellular Ca^2+^ entry (Figure 3C,D). To rule out the involvement of SMOCs in the Ca^2+^ response to GABA, we inhibited Transient Receptor Potential Vanilloid 4 (TRPV) channels, which are expressed in hCMEC/D3 cells and are sensitive to RN-1734 [57]. Appendix A shows that pre-treatment with RN-1734 (20 µM) did not affect GABA-evoked extracellular Ca^2+^ influx. Conversely, hCMEC/D3 cells express only very low levels of TRP Canonical 7 (TRPC7) channels [32], while they lack other TRPC channel isoforms that can support agonist-dependent Ca^2+^ influx in vascular endothelial cells, such as TRPC3 and TRPC6 [26,58]. Altogether, these findings indicate that SOCE supports GABA-induced intracellular Ca^2+^ signals in hCMEC/D3 cells.

### 3.4. InsP_3_ and Nicotinic Acid Adenine Dinucleotide Phosphate (NAADP) Trigger GABA-Induced Intracellular Ca^2+^ Release in hCMEC/D3 Cells

Growing evidence indicates that neurotransmitters and neuromodulators elicit intracellular Ca^2+^ release in the hCMEC/D3 cell line that is triggered by InsP_3_-induced ER Ca^2+^ mobilization through InsP_3_Rs and supported by NAADP-dependent lysosomal Ca^2+^ release via TPCs [35,36,37]. In accord, GABA-evoked intracellular Ca^2+^ mobilization was suppressed by blocking PLCβ activity with the aminosteroid U73122 (10 µM) [32,35,36,37] (Figure 4A) and by inhibiting InsP_3_Rs with the non-competitive antagonist 2-aminoethoxydiphenyl borate (2-APB; 50 µM) [32,35,36,37] (Figure 4A). Furthermore, the endogenous Ca^2+^ response to GABA was repressed by depleting the ER Ca^2+^ store with cyclopiazonic acid (CPA; 30 µM) (Figure 4B), which selectively affects the sarco-endoplasmic reticulum Ca^2+^ ATPase (SERCA) activity [32,35,36,37]. The statistical analysis of these experiments has been reported in Figure 4C. Lysosomal Ca^2+^ release via TPCs can recruit juxtaposed InsP_3_Rs through the mechanism of Ca^2+^-induced Ca^2+^ release, thereby triggering the Ca^2+^ response to extracellular stimuli in hCMEC/D3 cells [35,36,37], as well as in other endothelial cell types [59]. Figure 4D shows that GABA-induced intracellular Ca^2+^ release was abrogated by depleting the lysosomal Ca^2+^ pool with nigericin (50 µM), which acts as a H^+^/K^+^ antiporter and thereby dissipates the H^+^ gradient that maintains lysosomal Ca^2+^ refilling [60,61,62]. Moreover, the endogenous Ca^2+^ response to GABA (100 µM) was strongly inhibited by two specific NAADP antagonists, NED-19 (100 µM) (Figure 4E) and its chemically modified analogue, NED-K (100 µM) (Figure 4E) [60,63]. The analysis of these results has been illustrated in Figure 4F. Taken together, these findings demonstrate that InsP_3_-induced ER Ca^2+^ release and NAADP-evoked lysosomal Ca^2+^ discharge shape GABA-evoked intracellular Ca^2+^ mobilization in hCMEC/D3 cells.

### 3.5. GABA_A_ and GABA_B_ Receptors Mediate GABA-Induced Intracellular Ca^2+^ Signals in hCMEC/D3 Cells

Recent work showed that GABA_A_ receptors initiate the Ca^2+^ response to GABA in mouse cerebrovascular endothelial cells [17]. However, GABA_B_ receptors, which trigger an increase in [Ca^2+^]_i_ in both neurons [9,10] and astrocytes [12,13], were recently found to mediate intracellular Ca^2+^ signals also in human aortic endothelial cells (HAECs) [64]. To disentangle the GABA receptors involved in the endothelial Ca^2+^ signals (Figure 5A), we first exploited a battery of selective inhibitors of GABA_A_ and GABA_B_ receptors. Blocking GABA_A_ receptors with SR95531 (gabazine; 10 µM) strongly reduced the percentage of responding cells and significantly reduced the peak Ca^2+^ response in the minority of hCMEC/D3 cells displaying a detectable Ca^2+^ signal [2,22] (Figure 5B,D,E). The same inhibitory effects were achieved by selectively inhibiting GABA_B_ receptors with either saclofen (200 µM) or CGP35348 (100 µM) (Figure 5C–E). These data, therefore, demonstrate that both GABA_A_ and GABA_B_ receptors mediate the Ca^2+^ response to GABA in hCMEC/D3 cells.

To further corroborate these findings, we took advantage of two different agonists of GABA_A_ and GABA_B_ receptors, i.e., respectively, muscimol [17,22] and baclofen [12,13,22]. Figure 6A shows that both muscimol (15 µM) and baclofen (100 µM) induced a biphasic increase in [Ca^2+^]_i_ that closely resembled the Ca^2+^ response to GABA (100 µM). There was no significant difference in the percentage of responding cells (Figure 6B), while the amplitude of the peak Ca^2+^ response to GABA was significantly (*p* < 0.0001) higher as compared to both muscimol and baclofen (Figure 6C). The Ca^2+^ response to the combined application of muscimol (15 µM) and baclofen (100 µM) was also significantly (*p* < 0.0001) lower as compared to GABA (Figure 6). Nevertheless, these results confirm that both GABA_A_ and GABA_B_ contribute to triggering GABA-induced intracellular Ca^2+^ signals in the human cerebrovascular endothelial cell line hCMEC/D3.

### 3.6. Evidence That GABA_A_ and GABA_B_ Receptors Must Interact to Increase the [Ca^2+^]_i_

The mechanistic analysis described in Section 3.3 and Section 3.4 revealed that the Ca^2+^ response to GABA is triggered by endogenous Ca^2+^ release from non-acidic (ER) and acidic (lysosomes) Ca^2+^ stores and maintained over time by SOCE. It has long been known that GABA_B_ receptors are coupled to InsP_3_-induced ER Ca^2+^ mobilization and SOCE activation [7,8,9,10,12,13]. Interestingly, emerging evidence indicates that GABA_A_ receptors could signal an increase in [Ca^2+^]_i_ in a flux-independent manner [22,23,24], i.e., without the need for Cl^−^ fluxes, possibly via the metabotropic GABA_B_ receptors [22]. In agreement with these observations, planar patch-clamp recordings revealed that neither GABA (100 µM) (Figure 7A) nor muscimol (15 µM) (Figure 7B) were able to elicit detectable transmembrane currents in hCMEC/D3 cells. Furthermore, muscimol (15 µM) evoked robust intracellular Ca^2+^ mobilization (Figure 7C), which demonstrates that GABA_A_ receptor stimulation is able to mobilize the intracellular Ca^2+^ pool. In agreement with this observation, restoration of extracellular Ca^2+^ led to SOCE activation in 101 out of 147 hCMEC/D3 cells (Figure 7C). These findings suggest that the GABA_A_ subunits present in hCMEC/D3 cells assemble into a pentameric complex that is able to trigger intracellular Ca^2+^ signals by promoting intracellular Ca^2+^ release, but not to conduct sizeable Cl^−^ currents, as recently shown in mouse ciliated oviductal cells [22]. In accord, the Ca^2+^ response to muscimol (15 µM) was significantly inhibited by blocking GABA_A_ receptors with gabazine (10 µM) (Figure 7D,E). Notably, muscimol-evoked intracellular Ca^2+^ signals in hCMEC/D3 cells were also strongly reduced by inhibiting GABA_B_ receptors with either saclofen (200 µM) or CGP-35348 (100 µM) (Figure 7D,E). Therefore, GABA_A_ receptors require functional GABA_B_ receptors to signal the increase in [Ca^2+^]_i_ in a flux-independent manner. Similarly, the Ca^2+^ response to baclofen (100 µM) was sensitive to the pharmacological blockade of either GABA_A_ receptors with gabazine (10 µM) (Figure 7F,G), or GABA_B_ receptors with saclofen (200 µM) and CGP-35348 (100 µM) (Figure 7F,G). Therefore, the sequence of signaling events leading to GABA-induced intracellular Ca^2+^ signals in hCMEC/D3 cells requires the functional interaction between both GABA_A_ and GABA_B_ receptors, with GABA_A_ receptors operating in a non-canonical (i.e., flux-independent) manner via GABA_B_ receptors.

To further confirm this hypothesis, we pretreated hCMEC/D3 cells with pertussis toxin (PTx), which blocks the heterotrimeric G_i/o_ proteins that recruit PLCβ upon GABA_B_ receptor activation [10,12,22]. Figure 8 shows that PTx (1 ng/mL) significantly reduced the Ca^2+^ response to GABA (100 µM), muscimol (15 µM), or baclofen (100 µM). These findings confirm that GABA_B_ receptors signal the increase in [Ca^2+^]_i_ via G_i/o_ proteins and lend further support to the evidence that GABA_B_ receptors are also crucial to the onset of GABA_A_-receptor-mediated Ca^2+^ signals.

## 4. Discussion

Herein, we provided the first evidence that both GABA_A_ and GABA_B_ receptors mediate the Ca^2+^ response to the inhibitory neurotransmitter GABA in the widely employed human cerebrovascular endothelial cell line, hCMEC/D3. We further show that GABA_A_ receptors operate in a metabotropic, i.e., flux-independent, mode, to signal the downstream increase in [Ca^2+^]_i_ via the metabotropic GABA_B_ receptors. In line with this evidence, GABA_B_ receptors require functional GABA_A_ receptors to elevate the [Ca^2+^]_i_ in hCMEC/D3 cells. Unravelling the molecular mechanisms involved in endothelial GABA signaling will contribute to gather further insights into the mechanisms whereby endothelial GABAergic signaling regulates the NVU.

### 4.1. The Expression Profile of GABA_A_ and GABA_B_ Receptors in hCMEC/D3 Cells

The GABAergic innervation of intracortical microvessels by local GABA interneurons, rather than basal forebrain GABAergic terminals, has long been identified [15,65,66]. Subsequently, autoradiography with use of the GABA_A_ receptor agonist, muscimol, demonstrated specific binding sites in cerebral arterioles [67]. More recently, a thorough RT-PCR characterization showed that mouse brain cerebrovascular endothelial cells express the α1, α2, α6, β1, β2, β3, γ1, γ2 and γ3 subunits of GABA_A_ receptors [18]. A similar pattern of expression has been also detected in mouse embryonic forebrain endothelial cells [17]. In the present investigation, we found that hCMEC/D3 cells express the following GABA_A_ receptor subunits: α1 (also confirmed by immunoblotting), α5, β1, β2, and γ1. Therefore, human cerebrovascular endothelial cells possess all the three distinct GABA_A_ receptor subunits that can arrange into a pentameric ion channel with the likely stoichiometry 2α:2β:1γ [3,6]. Based upon their expression levels, GABA_A_ receptors in hCMEC/D3 cells are predicted to incorporate the α1, β1, and γ1 subunits. Surprisingly, human cerebrovascular endothelial cells do not present the GABA_A_ receptor β3 subunit, which is a crucial component of mouse GABA_A_ receptors [17,20,21]. The expression of the GABA_B_ receptor has hitherto been suggested only by an early study reporting on baclofen-induced nitric oxide (NO) release and collagen constriction in mouse cortical microvascular endothelial cells [19]. Herein, we provided the first evidence that both GABA_B1_ and GABA_B2_ subunits are expressed in hCMEC/D3 cells with GABA_B1_ showing enriched expression. Thus, a functional GABA_B_ receptor heterodimer can be assembled in human cerebrovascular endothelium.

### 4.2. GABA Induces Intracellular Ca^2+^ Signals in hCMEC/D3 Cells

GABA is emerging as a crucial mediator of neuro-to-vascular communication at the NVU [68]; mouse cerebrovascular endothelial cells express ionotropic GABA_A_ receptors that perceive GABA released during neuronal activity from inhibitory interneurons and trigger a signaling pathway that finely controls cerebral angiogenesis [17] and CBF [20]. It has been suggested that GABA activates GABA_A_ receptors to evoke intracellular Ca^2+^ signals [17,20], which could, in turn, drive endothelial cell proliferation and angiogenesis [25,26,69] and recruit endothelial nitric oxide synthase (eNOS) to produce NO, i.e., the most important vaso-relaxing mediator in the brain [27,28,31,68]. Nevertheless, GABA_A_ receptors are ionotropic receptors that are permeable to Cl^−^ and, therefore, are not expected to directly raise the [Ca^2+^]_i_ during GABAergic signaling [6]. The metabotropic GABA_B_ receptors have recently been shown to cause a transient increase in [Ca^2+^]_i_ in HAECs, but it is still unknown whether they are able to regulate the [Ca^2+^]_i_ also in cerebrovascular endothelial cells. We found that GABA evoked a biphasic Ca^2+^ response over a wide concentration range in hCMEC/D3 cells. The Ca^2+^ response to GABA was already detectable as concentrations as low as 1 pM and achieved the peak at 1 µM. The amplitude of the initial Ca^2+^ peak progressively decreased with further increases in agonist concentration. This dose–response relationship is quite different from the sigmoidal curve that has been described for GABA-evoked intracellular Ca^2+^ signals in neurons and astrocytes [70], in which the Ca^2+^ response is exclusively mediated by the metabotropic GABA_B_ receptors [9,10,11,12,13,70]. The evidence, which is further illustrated below, that the endothelial Ca^2+^ response to GABA is triggered by both GABA_A_ and GABA_B_ receptors could explain the peculiar profile of the dose–response relationship obtained in hCMEC/D3 cells. These preliminary observations confirmed that intracellular Ca^2+^ signaling could be instrumental for GABA to control endothelial cell functions [17,20]. A novel GABA biosensor based upon a dual-enzyme immobilization approach recently showed that, during neuronal activity, GABA concentration may transiently raise up to ≈80 µM in the cortex [47]. Ca^2+^ imaging recordings showed that 100 µM GABA elicits a robust biphasic elevation in [Ca^2+^]_i_ in hCMEC/D3 cells, and this concentration was used to unveil the underlying signaling pathways. Notably, GABA-induced intracellular Ca^2+^ waves have been observed in many cell types that do not belong to the NVU, including HAECs [64], human aortic smooth muscle cells [71], mouse embryonic stem cells [72], and the human breast cancer cell line, MCF7 [24].

### 4.3. The Complex Mechanisms of GABA-Induced Intracellular Ca^2+^ Signals in hCMEC/D3 Cells: InsP_3_Rs, TPCs and SOCE

Previous studies showed that, in neurons and astrocytes, the Ca^2+^ response to GABA was initiated by GABA_B_ receptors and comprised a rapid Ca^2+^ transient that is shaped by ER Ca^2+^ release through InsP_3_Rs followed by SOCE [9,10,11,12,13]. The PLCβ signaling pathway has been invoked also as molecular driver of GABA-induced intracellular Ca^2+^ signals outside the NVU [24,64,71,72]. A recent series of studies demonstrated that, in hCMEC/D3 cells, the Ca^2+^ response to neurotransmitters and neuromodulators, such as acetylcholine [32], ATP [33,34], glutamate [35,37], histamine [36] and arachidonic acid [57], is triggered by InsP_3_-evoked Ca^2+^ release from the ER and maintained over time by SOCE. In addition, NAADP-induced lysosomal Ca^2+^ discharge via TPCs can support InsP_3_-dependent ER Ca^2+^ release [59,61], thereby adding a further layer of complexity to the molecular mechanisms that pattern endothelial Ca^2+^ signals. Conversely, hCMEC/D3 cells lack ryanodine receptors [32], which may amplify InsP_3_-induced ER Ca^2+^ release through the Ca^2+^-induced Ca^2+^-release (CICR) mechanism in other endothelial cell types [73].

#### 4.3.1. InsP_3_Rs and TPCs

Herein, pharmacological manipulation of extracellular Ca^2+^ concentration confirmed that also the Ca^2+^ response to GABA was dependent upon intra- and extracellular Ca^2+^ sources. In the absence of external Ca^2+^, GABA evoked a smaller and transient elevation in [Ca^2+^]_i_, as previously reported in cortical neurons [9] and astrocytes [13], as well as in HAECs [64] and human aortic smooth muscle cells [71]. The GABA-evoked intracellular Ca^2+^ release in hCMEC/D3 cells was strongly impaired by blocking PLCβ activity with U73122, by inhibiting InsP_3_Rs with 2-APB, and by depleting ER Ca^2+^ content with CPA. As previously shown for acetylcholine [32], glutamate [35], ATP [33], and histamine [36], these findings convincingly indicate that InsP_3_-induced ER Ca^2+^ mobilization initiates the Ca^2+^ response to GABA in hCMEC/D3 cells. NAADP-evoked lysosomal Ca^2+^ discharge via TPCs is emerging as a crucial mechanism shaping endothelial Ca^2+^ signaling in peripheral vasculature [59,60,61,74,75,76,77]. Likewise, GABA-induced intracellular Ca^2+^ release was disrupted by pharmacologically emptying the lysosomal Ca^2+^ store with nigericin and by inhibiting TPCs with either NED-19 or NED-K. These results lend further support to the emerging notion that TPCs finely tune the Ca^2+^ response to extracellular stimulation not only in cerebrovascular endothelial cells [32,35,36,37,57], but also in neurons and astrocytes [78,79,80]. It has been suggested that lysosomal Ca^2+^ release through TPCs could trigger cytosolic CICR responses from the ER through InsP_3_Rs at juxtaposed ER–lysosome contact sites [61,62,63,81]. An alternative, but not mutually exclusive, mechanism whereby the lysosomal Ca^2+^ store could contribute to InsP_3_-driven Ca^2+^ signals is by refilling the ER with Ca^2+^ [82]. On the other hand, InsP_3_-induced Ca^2+^ release could induce the Ca^2+^-dependent production of NAADP [83] or favor lysosomal Ca^2+^ loading [84], which could activate TPC2 even in the absence of its ligand [85]. Although elucidating the Ca^2+^-dependent cross-talk between the ER and lysosomal Ca^2+^ stores in hCMEC/D3 cells is far beyond the scope of the present investigation, there is no doubt that both InsP_3_Rs and TPCs contribute to GABA-evoked intracellular Ca^2+^ release.

#### 4.3.2. SOCE

SOCE represents the Ca^2+^ entry pathway that mediates extracellular Ca^2+^ entry evoked by chemical cues in endothelial cells across the whole peripheral vasculature [25,86,87]. Likewise, also in hCMEC/D3 cells, SOCE sustains extracellular Ca^2+^ influx in response to acetylcholine [32], glutamate [35], and histamine [36]. In addition, SOCE can also be activated downstream of NMDARs, which signal in a flux-independent manner by recruiting the PLCβ signaling pathway [37]. Herein, we found that the re-addition of external Ca^2+^ after GABA-dependent depletion of the ER Ca^2+^ stores, and in the absence of the agonist, evoked robust extracellular Ca^2+^ influx. Under these conditions, GABA is no longer bound to its membrane receptors and, therefore, it is unlikely to stimulate the production of intracellular second messengers, such as arachidonic acid [57], which are able to gate Ca^2+^-permeable channels on the plasma membrane [32,35,38]. In agreement with these observations, blocking SOCE with either Pyr6 or BTP-2 significantly reduced or abolished GABA-evoked extracellular Ca^2+^ influx, whereas low micromolar doses of Gd^3+^ suppressed it. The different extents of SOCE inhibition between the pyrazole derivatives, Pyr6 and BTP-2, and the trivalent cation, Gd^3+^, could be due to their distinct mechanisms of action. In accord, while Gd^3+^ directly plugs the channel pore of Orai1 protein, Pyr6 and BTP-2 are likely to interfere with Orai1 recruitment by STIM1 [55,88,89]. Of note, the pharmacological blockade of TRPV4 channels, which are also expressed in hCMEC/D3 cells [57], with RN-1734 did not affect GABA-evoked Ca^2+^ entry. This finding, therefore, extends the repertoire of neurotransmitters that impinge on SOCE to generate long-lasting Ca^2+^ signals within the NVU and raises the question as to whether SOCE is recruited by GABA also in HAECs [64], human aortic smooth muscle cells [71], mouse embryonic stem cells [72], and MCF-7 breast cancer cells [24].

### 4.4. GABA_A_ and GABA_B_ Receptors Mediate GABA-Induced Intracellular Ca^2+^ Signals in hCMEC/D3 Cells

Although the metabotropic GABA_B_ receptors are known to induce intracellular Ca^2+^ signals both within [9,10,11] and outside [24,64,71,72] the NVU, the increase in the [Ca^2+^]_i_ whereby GABA regulates multiple endothelium-dependent functions in brain microvessels is mediated by the ionotropic GABA_A_ receptors [17,20]. GABA_A_ receptor activation by the selective agonist muscimol elicits an inward Cl^−^ current in mouse cerebrovascular endothelial cells, thereby elevating the [Ca^2+^]_i_ [17]. A potential explanation for this finding is that GABA-induced hyperpolarization enhances the driving-force sustaining the constitutive influx of Ca^2+^ occurring in these cells [90]. If this hypothesis holds true, muscimol should not increase the [Ca^2+^]_i_ in the absence of extracellular Ca^2+^.

Preliminary experiments revealed that both GABA_A_ and GABA_B_ receptors mediate the Ca^2+^ response to GABA. In accord, GABA-evoked intracellular Ca^2+^ signals were significantly reduced by inhibiting both the ionotropic GABA_A_ receptors with gabazine and the metabotropic GABA_B_ receptors with saclofen or CGP35348. Moreover, muscimol and baclofen, which, respectively, activate GABA_A_ and GABA_B_ receptors, also elicited a biphasic Ca^2+^ response in hCMEC/D3 cells. The following pieces of evidence support the notion that the ionotropic GABA_A_ receptor may signal the increase in [Ca^2+^]_i_ in a flux-independent (i.e., metabotropic) mode. First, whole-cell patch-clamp recordings showed that neither GABA nor muscimol evoked a sizeable membrane current in hCMEC/D3 cells. Second, the Ca^2+^ response to muscimol was sensitive to gabazine. Third, muscimol induced a robust Ca^2+^ signal also under 0Ca^2+^ conditions, which demonstrates that GABA_A_ receptors are able to mobilize the endogenous Ca^2+^ pool and thereby can operate in a metabotropic manner [91,92]. Intriguingly, a recent investigation revealed that, in hCMEC/D3 cells, the ionotropic NMDARs do not mediate detectable non-selective cation currents, but signal increases in [Ca^2+^]_i_ in a flux-independent manner by interacting with mGluR1 and mGluR5 [37]. In agreement with our observations, a body of studies has recently showed that also GABA_A_ receptors present metabotropic activity and can induce InsP_3_-dependent ER Ca^2+^ release in MCF-7 breast cancer cells [24] and rat cortical neurons [23]. Furthermore, it has been demonstrated that GABA_A_ receptors may transactivate the G_i/o_ protein coupled GABA_B_ receptors in a PTX-dependent manner and promote InsP_3_-dependent ER Ca^2+^ release in mice ciliated oviductal cells [22]. Intriguingly, also in these cells, GABA_A_ receptor stimulation with muscimol did not activate any inward Cl- current [22]. Likewise, we found that the Ca^2+^ response to direct GABA_A_ receptor stimulation with muscimol was hampered by blocking GABA_B_ receptor signaling with either saclofen, CGP35348 or PTx. Unlike the oviduct [22], however, the Ca^2+^ response to baclofen was in turn inhibited by blocking GABA_A_ receptors with gabazine, as well as by preventing G_i/o_ protein activation with PTx.

These findings shed light upon an unusual mode of GABAergic signaling in the cerebrovascular endothelium. The Ca^2+^ response to GABA, which underpins GABA-induced cerebral angiogenesis [17] and the GABA-induced local increase in CBF [20], requires the functional interaction between GABA_A_ and GABA_B_ receptors. GABA_A_ receptors can engage metabolic signaling via GABA_B_ receptors, and GABA_B_ receptors fail to generate a full Ca^2+^ signal if GABA_A_ receptors are inhibited. In line with these findings, GABA_A_ and GABA_B1_ receptors have been shown to physically interact in mice ciliated oviductal cells [22] and in rat brain lysates [93], and to co-localize at multiple synaptic and extra-synaptic sites in the brain [94]. Future work will seek to understand the molecular underpinnings of the functional interaction between GABA_A_ and GABA_B_ receptors in hCMEC/D3 cells. The preliminary evidence that muscimol and baclofen do not exert an additive effect (described in Figure 6C), however, strongly suggests that the PLCβ signaling pathway is engaged by only one GABA receptor isoform, which is likely to be GABA_B_ [7,8]. This model is supported by the evidence that PTx also attenuates muscimol-evoked intracellular Ca^2+^ signals. Since the Ca^2+^ response to the mixture of muscimol and baclofen is lower as compared to GABA alone, we hypothesize that the two GABA receptor isoforms must somehow be “synchronized” to interact and elicit the full Ca^2+^ response and that this requires the presence of the physiological agonist.

## 5. Conclusions

Herein, we unveiled the complex signaling pathway whereby the inhibitory neurotransmitter GABA can induce an increase in [Ca^2+^]_i_ in human cerebrovascular endothelial cells. The Ca^2+^ response to GABA is triggered by InsP_3_-induced ER Ca^2+^ release and NAADP-dependent lysosomal Ca^2+^ mobilization, whereas it is mainly maintained by SOCE (Figure 9). Both GABA_A_ and GABA_B_ receptors support GABA-evoked intracellular Ca^2+^ signals (Figure 9). The ionotropic GABA_A_ receptors signal in a flux-independent manner via the metabotropic GABA_B_ receptors. Likewise, the full Ca^2+^ response to GABA_B_ receptor stimulation requires functional GABA_A_ receptors. Endothelial Ca^2+^ signals finely tune a myriad of vascular functions, including those regulated by GABA, such as angiogenesis and CBF control via NO release. Therefore, this study sheds novel light on the molecular mechanisms by which GABA controls endothelial signaling at the NVU.

## Figures and Tables

**Figure 1 cells-11-03860-f001:**
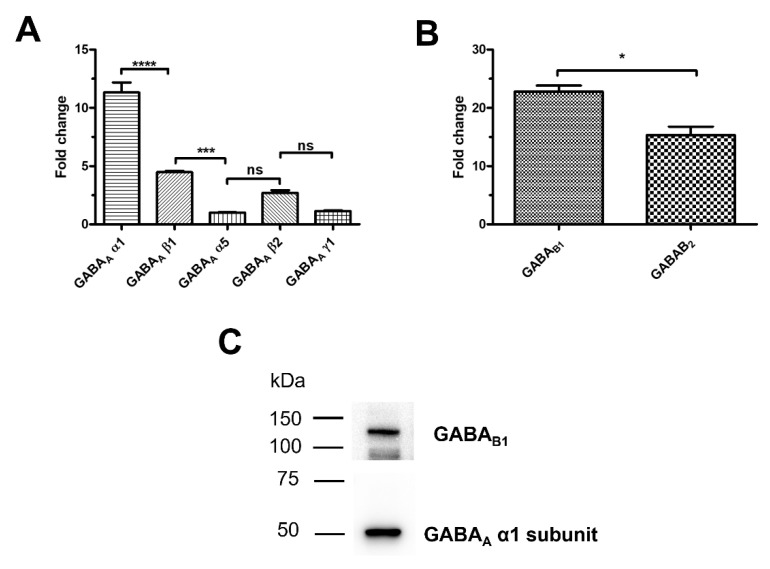
GABA_A_ and GABA_B_ receptors are expressed in hCMEC/D3 cells. (**A**), a panel of GABA_A_ receptor subunit transcripts are expressed in hCMEC/D3 cells based on qRT-PCR data. **** indicate *p* < 0.0001 *** indicate *p* < 0.001 (One-way ANOVA followed by the post-hoc Bonferroni test). NS indicates not significant. (**B**), both GABA_B1_ and GABA_B2_ receptor subunit transcripts are expressed in hCMEC/D3 cells. * indicates *p* < 0.05 (Student’s *t*-test). (**C**), representative immunoblots showing that both GABA_A_ α1 and GABA_B1_ subunits are expressed at protein level. qRT-PCR and immunoblotting analyses were carried out on three distinct biological replicates.

**Figure 2 cells-11-03860-f002:**
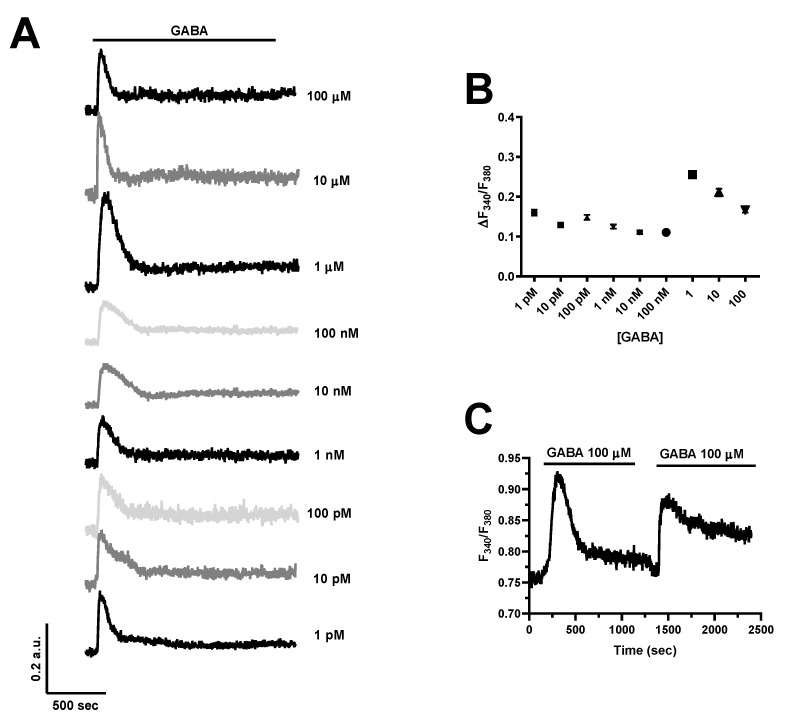
GABA evokes a dose–response increase in [Ca^2+^]_i_ in hCMEC/D3 cells. (**A**), the inhibitory neurotransmitter, GABA, elicits a dose-dependent elevation in [Ca^2+^]_i_, which remains relatively stable between 1 pM and 100 nM and achieves its peak at 1 µM. In this and the following figures, the black bar above the Ca^2+^ tracings indicates the time of agonist addition. (**B**), mean ± SE of the amplitude of the initial Ca^2+^ peak evoked by GABA at each agonist concentration ([GABA]). The number of analyzed cells ranges between 103 and 117 from three independent experiments for each dose. (**C**), GABA (100 μM) elicits an additional increase in [Ca^2+^]_i_ upon 3 min washout, which indicates that GABA receptors do not desensitize at this agonist concentration.

**Figure 3 cells-11-03860-f003:**
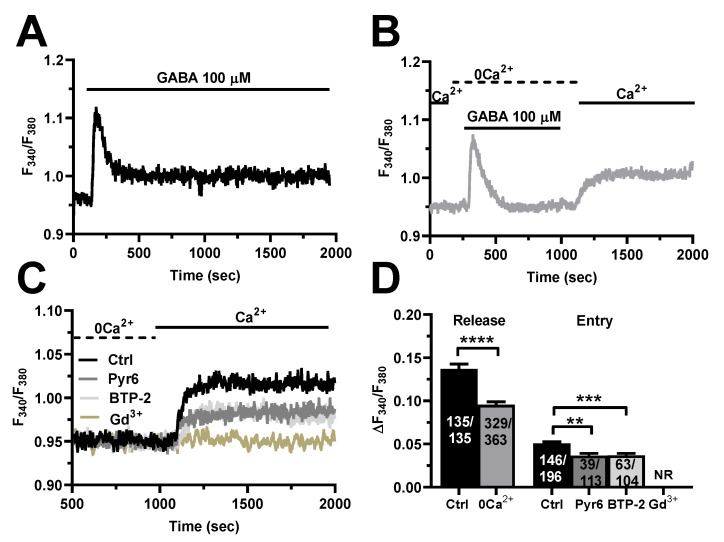
GABA evokes intracellular Ca^2+^ release and SOCE activation in hCMEC/D3 cells. (**A**), representative tracing of the biphasic Ca^2+^ response induced by GABA (100 µM) in the presence of extracellular Ca^2+^ (Ctrl). (**B**), representative tracing of the two distinct components of GABA-evoked intracellular Ca^2+^ signals in hCMEC/D3 cells. GABA induced a transient increase in [Ca^2+^]_i_ in the absence of extracellular Ca^2+^ (0Ca^2+^), which is due to endogenous Ca^2+^ mobilization. Restoration of extracellular Ca^2+^ (1.5 mM) after removal of the agonist induced a second increase in [Ca^2+^]_i_, which is indicative of SOCE activation. (**C**), GABA-evoked extracellular Ca^2+^ entry was significantly reduced in the presence of Pyr6 (10 µM, 10 min), BTP-2 (20 µM, 10 min), or Gd^3+^ (20 µM, 10 min), three specific SOCE inhibitors. Intracellular Ca^2+^ release is not shown. (**D**), mean ± SE of the amplitude of GABA-evoked Ca^2+^ peaks in the presence (Ctrl) and absence (0Ca^2+^) of extracellular Ca^2+^. **** indicate *p* < 0.0001 (Student’s *t*-test). Mean ± SE of the amplitude of GABA-evoked extracellular Ca^2+^ entry in the absence (Ctrl) and presence of Pyr6, BTP-2 or Gd^3+^. *** indicate *p* < 0.001 and ** indicate *p* < 0.01 (one-way ANOVA followed by the post-hoc Dunnett’s test). NR indicates “No Response” as evaluated in 89 cells from three different experiments.

**Figure 4 cells-11-03860-f004:**
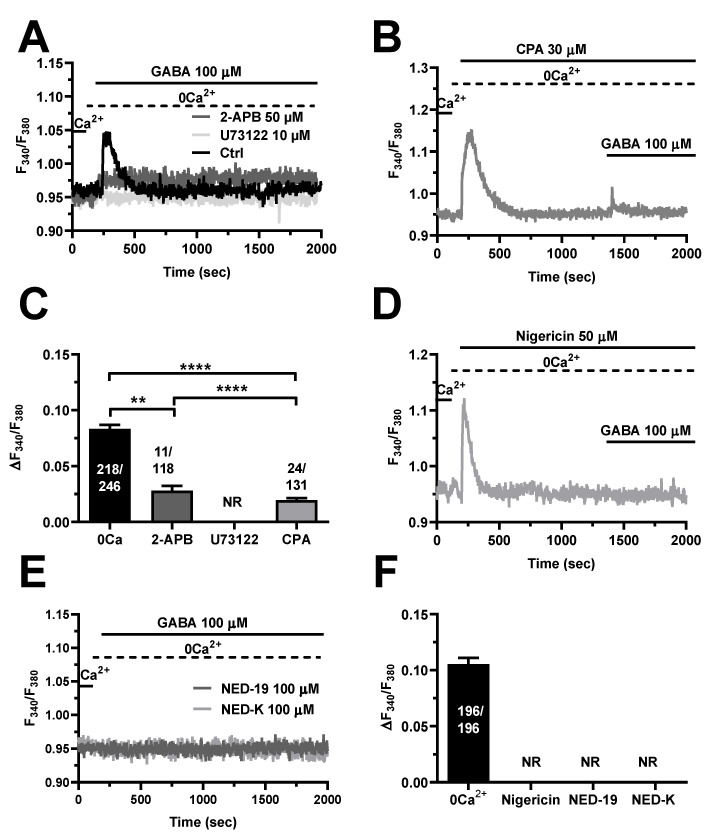
InsP_3_ and NAADP mediate GABA-evoked intracellular Ca^2+^ release in hCMEC/D3 cells. (**A**), representative tracings of GABA-evoked intracellular Ca^2+^ release in the absence (Ctrl) and presence of U73122 (10 µM, 10 min) or 2-APB (50 µM, 30 min), which, respectively, inhibit PLC and InsP_3_Rs. GABA was administered at 100 µM. (**B**), CPA (30 µM), a selective blocker of SERCA activity, induced a transient elevation in [Ca^2+^]_i_ under 0Ca^2+^ conditions due to passive ER Ca^2+^ efflux. The subsequent addition of GABA (100 µM) induced only a small Ca^2+^ transient, which was due to the lower ER Ca^2+^ content. (**C**), mean ± SE of the peak amplitude of GABA-evoked intracellular Ca^2+^ release under the designated treatments. **** indicate *p* < 0.0001, ** indicate *p* < 0.001 (One-way ANOVA followed by the post-hoc Bonferroni test). NR indicates “No Response” as evaluated in 102 cells from three different experiments. (**D**), depleting the lysosomal Ca^2+^ pool with the selective H^+^/K^+^ antiporter, nigericin (50 µM), caused a transient elevation in [Ca^2+^]_i_. The following addition of GABA (100 µM) failed to elicit a detectable Ca^2+^ signal. (**E**), inhibiting NAADP signaling with NED-19 (100 µM) or NED-K (100 µM) abolished the intracellular Ca^2+^ mobilization induced by GABA (100 µM). (**F**), mean ± SE of the peak amplitude of GABA-evoked intracellular Ca^2+^ release under the designated treatments. NR indicates “No Response” as evaluated in 104 cells treated with nigericin, with 101 cells treated with NED-19 and 99 cells treated with NED-K.

**Figure 5 cells-11-03860-f005:**
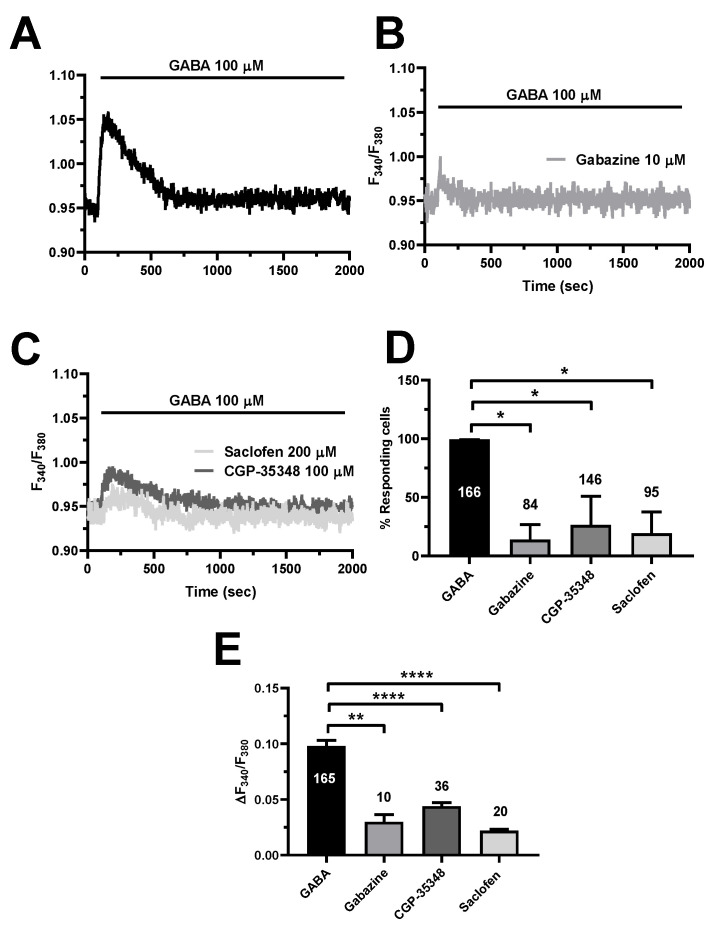
The pharmacological blockade of GABA_A_ and GABA_B_ receptors inhibit GABA-evoked intracellular Ca^2+^ signals in hCMEC/D3 cells. (**A**), representative tracing of the Ca^2+^ response induced by GABA (100 µM) under control (Ctrl) conditions. (**B**), blocking GABA_A_ receptors with gabazine (10 µM, 5 min) significantly reduced GABA-evoked intracellular Ca^2+^ signals. (**C**), blocking GABA_B_ receptors with either saclofen (200 µM, 5 min) or CGP-35348 (100 µM, 5 min) significantly reduced GABA-evoked intracellular Ca^2+^ signals. (**D**), mean ± SE of the percentage of hCMEC/D3 cells responding to GABA under the designated treatments. * indicates *p* < 0.05 (one-way ANOVA followed by the post-hoc Dunnett’s test). (**E**), mean ± SE of the amplitude of the peak Ca^2+^ response to GABA under the designated treatments. **** indicates *p* < 0.0001, ** indicates *p* < 0.01 (one-way ANOVA followed by the post-hoc Dunnett’s test).

**Figure 6 cells-11-03860-f006:**
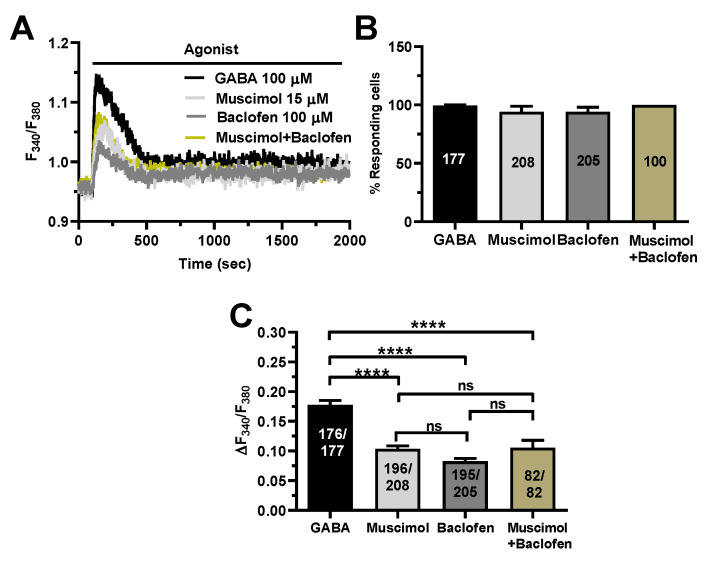
Selective stimulation of GABA_A_ and GABA_B_ receptors induce intracellular Ca^2+^ signals in hCMEC/D3 cells. (**A**), representative tracings of the biphasic Ca^2+^ signals induced by GABA (100 µM), the GABA_A_ receptor agonist, muscimol (15 µM), the GABA_B_ receptor agonist, baclofen (100 µM), and muscimol (15 µM) + baclofen (100 µM), in hCMEC/D3 cells. (**B**), mean ± SE of the percentage of hCMEC/D3 cells responding to GABA, muscimol and baclofen. (**C**), mean ± SE of the amplitude of the peak Ca^2+^ response to GABA, muscimol and baclofen. **** indicate *p* < 0.0001 (one-way ANOVA followed by the post-hoc Bonferroni test). NS indicates not significant.

**Figure 7 cells-11-03860-f007:**
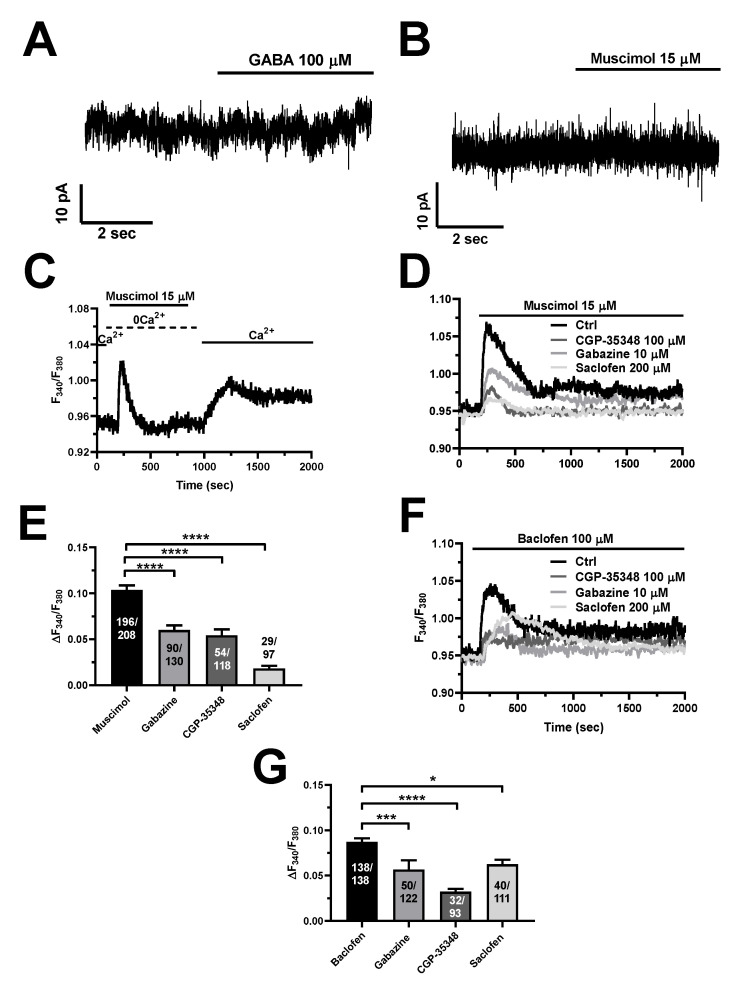
GABA-evoked intracellular Ca^2+^ signals require the functional interaction between GABA_A_ and GABA_B_ receptors in hCMEC/D3 cells. Planar whole-cell patch-clamp recordings revealed that neither GABA (100 µM) (**A**) nor muscimol (15 µM) (**B**) induced any inward current at a V_h_ = −70 mV. (**C**), the Ca^2+^ add-back protocol showed that muscimol (15 µM) was able to mobilize the intracellular Ca^2+^ pool under 0Ca^2+^ conditions and to activate SOCE on restoration of extracellular Ca^2+^ levels (1.5 mM) in the absence of the agonist. (**D**), the Ca^2+^ response to muscimol (15 µM) was reduced by blocking either GABA_A_ receptors with gabazine (10 µM, 5 min), or GABA_B_ receptors with saclofen (200 µM, 5 min) or CGP-35348 (100 µM, 5 min). (**E**), mean ± SE of the amplitude of the peak Ca^2+^ response to muscimol in the absence (Ctrl) or in the presence of gabazine, CGP-35348 or saclofen. **** indicate *p* < 0.0001 (one-way ANOVA followed by the post-hoc Dunnett’s test). (**F**), the Ca^2+^ response to baclofen (100 µM) was reduced by blocking either GABA_A_ receptors with gabazine (10 µM, 5 min), or GABA_B_ receptors with saclofen (200 µM, 5 min) or CGP-35348 (100 µM, 5 min). (**G**), mean ± SE of the amplitude of the peak Ca^2+^ response to baclofen in the absence (Ctrl) or in the presence of gabazine, CGP-35348 or saclofen. **** indicate *p* < 0.0001, *** indicate *p* < 0.001, * indicates *p* < 0.05 (one-way ANOVA followed by the post-hoc Dunnett’s test).

**Figure 8 cells-11-03860-f008:**
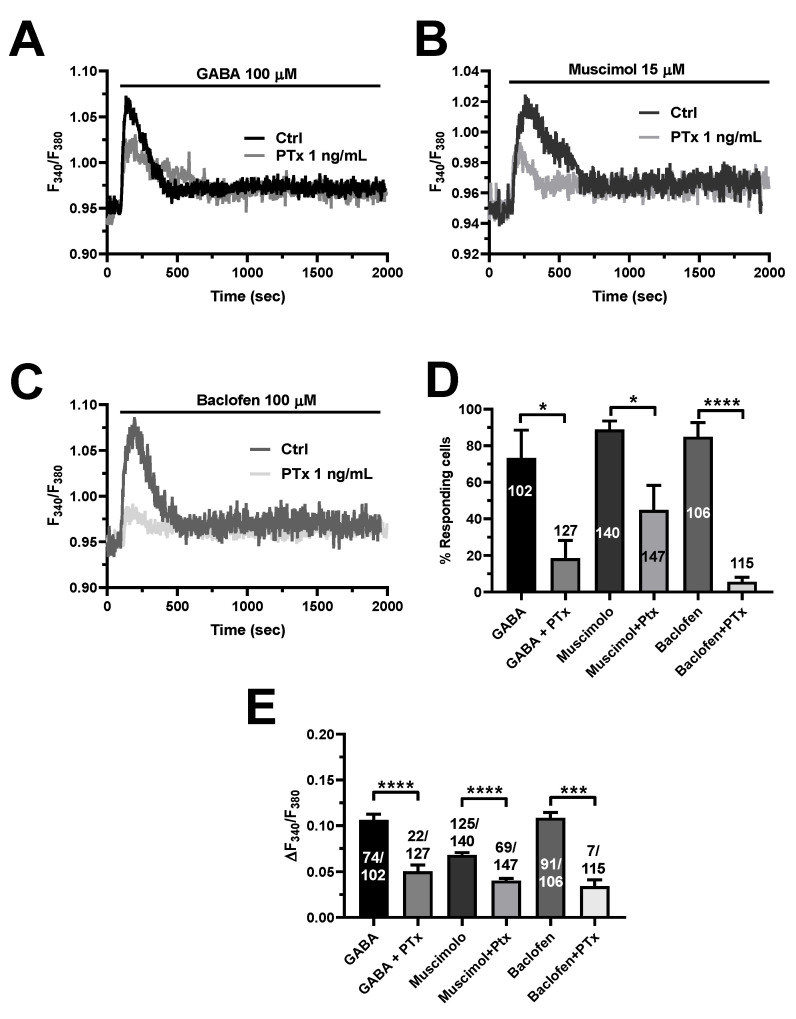
Selective inhibition of heterotrimeric G_i/o_ proteins with pertussis toxin (PTX) reduces GABA_A-_ and GABA_B_-induced intracellular Ca^2+^ signals in hCMEC/D3 cells. (**A**), representative tracings of the biphasic Ca^2+^ signals induced by GABA (100 µM), in the presence and in the absence of PTx (1 ng/mL). (**B**), Representative traces of the Ca^2+^ response induced by the GABA_A_ receptor agonist, muscimol (15 µM), in the presence and in the absence of PTx (1 ng/mL). (**C**), Representative Ca^2+^ traces evoked by the GABA_B_ receptor agonist, baclofen (100 µM), in the presence and in the absence of PTx (1 ng/mL). (**D**), mean ± SE of the percentage of hCMEC/D3 cells responding under the designated treatments. **** indicate *p* < 0.0001, * indicates *p* < 0.05 (Student’s *t*-test). (**E**), mean ± SE of the amplitude of the peak Ca^2+^ response under the designated treatments. **** indicate *p* < 0.0001, *** indicates *p* < 0.001 (Student’s *t*-test).

**Figure 9 cells-11-03860-f009:**
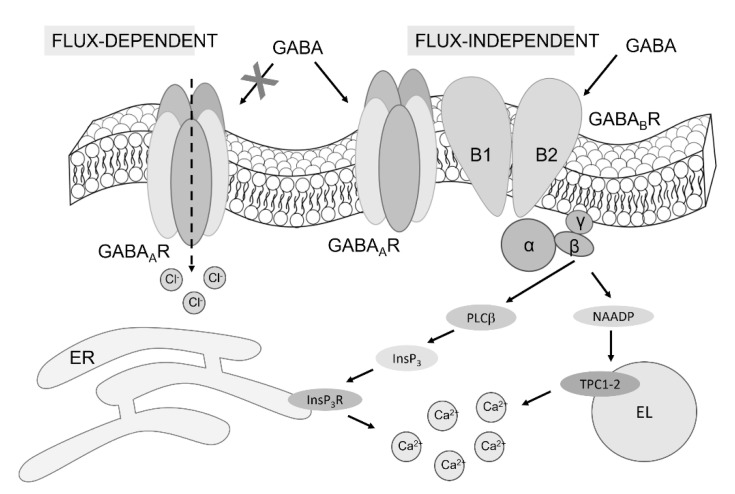
The mechanism of GABA-evoked intracellular Ca^2+^ signals in hCMEC/D3 cells. The evidence presented in this investigation demonstrates that both GABA_A_ and GABA_B_ receptors support the Ca^2+^ response to GABA in hCMEC/D3 cells. GABA_A_ receptors are unlikely to signal the increase in [Ca^2+^]_i_ in a flux-dependent manner, whereas they can operate in a metabotropic manner by interacting with GABA_B_ receptors. This interaction leads to InsP_3_-induced Ca^2+^ mobilization from the ER and NAADP-induced EL Ca^2+^ release through TPCs. The following reduction in ER Ca^2+^ concentrations, in turn, activates SOCE (not illustrated here).

## Data Availability

Data supporting reported results can be obtained by the corresponding author upon reasonable request.

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
