# Peer review of "GABAA and GABAB Receptors Mediate GABA-Induced Intracellular Ca2+ Signals in Human Brain Microvascular Endothelial Cells"

_cells, 2022, doi:10.3390/cells11233860_

Round 1

Reviewer 1 Report

1. Abstract needs to be re-written in a similar format as shown by current/recent articles in Cells. Methods should be removed as these are standard, not novel.

2. The word impinge (line 112-113) generally has a negative connotation. Do the authors mean to say that one receptor type is predominant over the other, depending on the stimulus, to generate a full response? Or both types are required for a full response?

3. Section 2.7 in Materials and Methods - Dunnett's was used in Figures 1 and 3.  Similarly, for PCR (see section 2.4), Tukey's was used. Please include both Tukey's Dunnetts's in the methods, and explain when each test (Tukey's, Dunnett's or Bonferroni) is used for post-hoc tests.

4. Figure 1: In the methods, the authors said that they used Tukey's and yet in the figure legend, different tests were described? Why is this so?  Furthermore, Tukey's post host test is used when comparing each mean with every other mean in the same dataset. I don't think that is what the authors are trying to achieve here.

Fig 1A: Use the same statistical test for all in the same data set. Do not use different tests on the same data set. Bonferroni should be used since multiple comparisons between means are made. Dunnett's post hoc test is used when comparing every mean with a control mean.

Fig 1B: What test was used for this - Student's T-test?

qRT-PCR and Western blots - how many experiments? 

4. Figure 2 - The peak for 100pm looks almost as big as the peak for 10uM, and also bigger than for 100uM. However, the calculated peak value (Fig 2B) for 100pm is much less. Is the trace used in Fig 2A correct?

5. Section 3.3 - Synta-66 has been shown to have higher selectivity in inhibiting Orai1 than BTP-2. The authors should have used Synta-66 or even knockdown Orai1 protein expression using siRNA to evaluate the contribution of Orai1 to GABA-induced calcium responses.

Since Pyr6 and BTP-2 did not completely knockdown GABA-evoked extracellular calcium influx, did the authors look at other channels that also contribute to SOCE? In general, endothelial cells have been shown to express TRPC, TRPV and TRPM channels.

6. Figure 3 - Reorganize the figure to fill up the space allocated, as there is a fair bit of whitespace. Specify how many cells were used and the number of experiments conducted.

7. Figure 4 - Reorganize the figure to fill up the space allocated, as there is a fair bit of whitespace. Specify how many cells were used and the number of experiments conducted. Also, Bonferroni should be used, not Dunnett's. 

Fig 4A: Show the control trace for GABA stimulation in this figure. 

Fig 4F: Explain what NR means in the figure legend.

8. Figure 5 - One-way Anova should not be used with percentage values - chi-square should be used. Reorganize the figure to fill up the space allocated, as there is a fair bit of whitespace. Specify how many cells were used and the number of experiments conducted.

9. Paragraph ending on line 393 - Left unsaid is the significant difference between muscimol (type A) and baclofen (type B) responses.  The authors should expand (in the discussion) on why the responses were smaller despite the % responding cells being equivalent.

Also, since both GABA types A and B receptors are required for a full calcium response, the authors should add both muscimol and baclofen together to check if their effects are additive.

10. Figure 6 - Bonferonni is the test to use, not Dunnett's. Also, specify how many cells were used and the number of experiments conducted.

11. Figure 7 - Specify how many cells were used and the number of experiments conducted.

12. Figure 8 - Specify how many cells were used and the number of experiments conducted.

Fig 8D: Unpaired Student' t test is not the right test for percentages. 

Fig. 8E: Please double check if this is the right test when presenting the data in this manner.

13. Section 4.3.2 - The authors should discuss and explain why Pyr6 and BTP-2 only blocked about 50% of GABA-evoked calcium influx. What other channels may be involved?

14. The authors should also specify, in the figure legend, whether the calcium traces shown are representative of  all cells in their experiments.

Reviewer 2 Report

In this manuscript the Authors evaluated the potential role of  GABA receptors stimulation on Ca2+ level in hCMEC/D3, the human cerebrovascular endothelial cell line. Pharmacological manipulation showed that GABA-induced intracellular Ca2+ release was mediated by ER Ca2+ mobilization through InsP3 receptors and NAADP-dependent lysosomal Ca2+ mobilization  via TPC1-2, whereas Ca2+ entry was mediated by SOCE. The Authors proposed the mechanism by which ionotropic GABAA and metabotropic GABAB receptors were both required to trigger the endothelial Ca2+ response. In addition, this study provided the first evidence that both GABAB1 and GABAB2 subunits are expressed in hCMEC/D3 cells. A study extends the previous works of this team, which have shown that several neurotransmitters, including acetylcholine, glutamate, ATP, histamine and arachidonic acid can act in hCMEC/D3 cells using similar pathway, ie. bind to the specific G-protein coupled receptors and by subsequent stimulation of PLCβ cleave PIP2 into InsP3 and DAG. In turn, InsP3 releases Ca2+ from the ER by InsP3 receptors. The presentation of the data is clear and the Author`s conclusion appeared to be valid. In my opinion the work should be accepted with only one remark. Intracellular Ca2+ is regulated through cooperation of various Ca2+ pumps and Ca2+ channels and one type of them are ryanodine receptors expressed in ER. Since the Authors have shown previously a lack of these receptors in hCMEC/D3 cells, it should be worth mentioning in the discussion.

Round 2

Reviewer 1 Report

1. Figure 1 - please also specify that ns indicates not significant in the legend. 

2.  Line 305 - I think the authors meant Figures 3C-D as there is no E. Please rectify this.

3.  I find it very interesting that no additive effects were observed when muscimol and baclofen are used together, given that both type GABAA and GABAB receptors are required to mediate GABA responses. I had expected a bigger response when used together. So, this is an interesting conundrum to ponder in future studies. 

Author Response

Dear Reviewer #1,

We are gratefully thankful for your additional comments on our manuscript entitled: “GABAA and GABAB receptors mediate GABA-induced intracellular Ca2+ signals in human brain microvascular endothelial cells” for publication as Research Article in Cells – Special Issue Cell Calcium across the Phylogenetic Tree: From Physiological Signaling to Pathogenic Mechanisms.

We amended the manuscript according to your suggestions.

More specifically:

1) Figure 1 - please also specify that ns indicates not significant in the legend.

We specified that NS indicates not significant in the legend.

2) Line 305 - I think the authors meant Figures 3C-D as there is no E. Please rectify this.

We thank the Referee for noticing this typo, that we amended.

3) I find it very interesting that no additive effects were observed when muscimol and baclofen are used together, given that both type GABAA and GABAB receptors are required to mediate GABA responses. I had expected a bigger response when used together. So, this is an interesting conundrum to ponder in future studies.

I do agree with the Referee. I have the feeling that ionotropic receptors might interact with their metabotropic counterparts more strongly than we previously thought. We have recently published that NMDA receptors cause Ca2+ release from ER and lysosomes by interacting with mGluR1 and mGluR5. The Valverde group demonstrated that GABAA receptors use GABAB receptors to induce ER Ca2+ release in mouse oviduct (PMID: 30108184; we have quoted this paper). So, I guess that future will reserve many surprises. Molecular studies are certainly required to deepen this issue and I hope we will find coworkers in Pavia willing to help us.

Once again, we do thank the Referee for the careful evaluation of the manuscript and her/his insightful comments. We do hope that you will now regard our manuscript suitable for publication on Cells.

Sincerely,

Francesco Moccia
